# Personalized tDCS for Focal Epilepsy—A Narrative Review: A Data-Driven Workflow Based on Imaging and EEG Data

**DOI:** 10.3390/brainsci12050610

**Published:** 2022-05-07

**Authors:** Steven Beumer, Paul Boon, Debby C. W. Klooster, Raymond van Ee, Evelien Carrette, Maarten M. Paulides, Rob M. C. Mestrom

**Affiliations:** 1Department of Electrical Engineering, University of Technology Eindhoven, P.O. Box 513, 5600 MB Eindhoven, The Netherlands; paul.boon@uzgent.be (P.B.); debby.klooster@ugent.be (D.C.W.K.); evelien.carrette@uzgent.be (E.C.); m.m.paulides@tue.nl (M.M.P.); r.m.c.mestrom@tue.nl (R.M.C.M.); 2Department of Neurology, Ghent University Hospital, Corneel Heymanslaan 10, 9000 Ghent, Belgium; 3Philips Research Eindhoven, High Tech Campus 34, 5656 AE Eindhoven, The Netherlands; raymond.van.ee@philips.com; 4Department of Radiation Oncology, Erasmus Medical Center Cancer Institute, Burgemeester Oudlaan 50, 3062 PA Rotterdam, The Netherlands

**Keywords:** neurostimulation, personalized, forward modeling, inverse modeling, transcranial electric stimulation, clinical outcome

## Abstract

Conventional transcranial electric stimulation(tES) using standard anatomical positions for the electrodes and standard stimulation currents is frequently not sufficiently selective in targeting and reaching specific brain locations, leading to suboptimal application of electric fields. Recent advancements in in vivo electric field characterization may enable clinical researchers to derive better relationships between the electric field strength and the clinical results. Subject-specific electric field simulations could lead to improved electrode placement and more efficient treatments. Through this narrative review, we present a processing workflow to personalize tES for focal epilepsy, for which there is a clear cortical target to stimulate. The workflow utilizes clinical imaging and electroencephalography data and enables us to relate the simulated fields to clinical outcomes. We review and analyze the relevant literature for the processing steps in the workflow, which are the following: tissue segmentation, source localization, and stimulation optimization. In addition, we identify shortcomings and ongoing trends with regard to, for example, segmentation quality and tissue conductivity measurements. The presented processing steps result in personalized tES based on metrics like focality and field strength, which allow for correlation with clinical outcomes.

## 1. Introduction

Neuromodulation concerns influencing neurons by external stimuli to modulate brain function. This field of research has seen a lot of promising results in recent years for treating numerous diseases and for improving the quality of life in highly prevalent brain conditions, such as Parkinson’s disease, spinal cord injuries, epilepsy, depression, and many more, which may be very difficult to treat using conventional medicine [1].

Neurostimulation is generally understood to be the form of neuromodulation that uses electricity to influence brain activity. The field of neurostimulation is a broad and expanding field in both clinical and technical research with a variety of techniques currently available, like deep brain stimulation (DBS), transcranial magnetic stimulation (TMS), transcranial electric stimulation (tES), or vagus nerve stimulation (VNS) [1]. Neurostimulation has successfully been applied to treat several disorders and is on its way to expanding its clinical approval. For example, repetitive transcranial magnetic stimulation (rTMS) is approved in the United States to treat depression [2], and DBS is already used for treating Parkinson’s disease [3], epilepsy [4], and obsessive-compulsive disorder (OCD) [5]. Since these methods rely on currents, voltages, and subsequently electromagnetic fields, it is evident that modeling these fields is crucial for assessing safety aspects and, possibly, the expected treatment outcome. In the case of tES, where individual anatomy plays a significant role, computational modeling of the electric fields inside the brain of an individual patient has found increasing interest, especially for the correlation of the field intensity in the region of interest (ROI) with the clinical treatment outcome. This can help predict how the stimulation should take place and what the outcome will be, reducing the current heterogeneity in tES treatment outcomes [6,7].

To accomodate tES treatments that can be personalized, we propose a data-driven workflow to obtain the stimulation focus and personalized electrode locations in a more rigorous and standardized way. With respect to electric field modeling and customizing tES treatments to steer stimulation, the research field of tES is rapidly advancing, as can be seen by the numerous recent publications. For our workflow, we provide a technical overview of the most recent findings in the literature that are used for personalization of the electrode montage during treatments. We do this for the case of personalizing tES for focal epilepsy. Starting from imaging data and optional electroencephalography (EEG) data, we review the available techniques for the necessary steps toward a personalized stimulation protocol, while taking into account uncertainties in the source maps and tissue parameters throughout the approach.

## 2. Rationale for Personalization: tES to Treat Focal Epilepsy

Epilepsy is a neurological disorder that has an estimated lifetime prevalence of 7.6 per 1000 people in a population [8]. It has a high degree of heterogeneity in its causes, symptoms, and severity, and the impact of epilepsy is substantial and broad. In addition to seizures, it may cause cognitive problems and psychological complaints, and it has negative socioeconomical consequences. This puts a heavy burden on individual health and quality of life, as well as on society. The traditional way of treating seizures is through anti-epileptic drugs (AEDs). In spite of adequately dosed AEDs, one out of three patients do not become seizure-free after using several (or combinations of) AEDs. These patients are called medically refractory patients [9]. Currently, the next step for refractory patients is to investigate whether they can be treated by resecting a small part of the brain that contains the epileptic focus, or the so-called epileptogenic zone [10]. This epileptogenic zone can be delineated by extensive pre-surgical evaluation, which combines the results of several techniques such as video EEG monitoring, magnetic resonance imaging (MRI), and positron emission tomography (PET) scans. Video EEG monitoring is being performed to record seizures via EEG and video simultaneously to identify the so-called ictal onset zone and symptomatic zone based on semiology. In addition, an MRI scan (often 3T) takes place to find an epileptic lesion congruent with the results of the other techniques. Analysis of interictal epileptiform discharges during EEG or magnetoencephalography (MEG) recordings can be used to find the location of the so-called irritative zone, which is the location where such interictal discharges are generated. Based on these combined noninvasive results, a hypothesis on the epileptogenic zone can guide epilepsy surgery, or it may need to be supplemented with intracranial recordings using subdural electrodes with or without depth electrodes [11].

Surgery is, obviously, an invasive step in treatment and not an option for all patients due to overlap with the eloquent cortex (i.e., specific brain areas that have direct control over specific brain functions, like motor functions, memory, and language) [12]. Less invasive or noninvasive techniques are also possible and would be preferable when applicable. A promising technique to treat epilepsy is neurostimulation [13,14,15,16], which could be in the form of VNS or DBS. TMS can also be used to treat epilepsy, although evidence of the efficacy for this method is low [17]. Over the past two decades, the noninvasive technique of tES, and specifically transcranial direct current stimulation (tDCS), has received increasing attention [18] for treating epilepsy.

This technique is, in essence, nothing more than the application of a current with a specified waveform to the scalp to induce a neuromodulatory effect. The waveform that is applied defines the type of stimulation. Apart from a constant current protocol (tDCS), alternating current protocols (tACS) and random noise current protocols (tRNS) exist.

The physiology of tDCS has been elaborated upon extensively in [19,20]. Further technical and safety aspects can be found in [21]. A review on the technical aspects like equipment, modeling, and stimulation protocols is given in [22]. The main hypothesis of the working principle is that the induced electric fields, which are too weak to directly invoke action potentials, can modulate the firing rate by causing a sub-threshold polarization effect on the neuron membrane potential. Lafon et al. used modeling combined with experiments on rat hippocampal brain slices to prove that this kind of stimulation modulates the firing rate in individual neurons [23]. A review of further advances from in vitro and in vivo models is given in [24].

The goal of tES for epilepsy is to decrease the excitability of the neurons in the targeted epileptogenic zone or in a so-called relay station (e.g., the thalamus) to lower the chances of hyper-excitability causing seizures. Although the exact mechanism of action is unknown, tDCS is believed to act mainly on the pyramidal cells in the cortex [20]. These are influenced mostly by the component of the electric fields normal to the cortex during neurostimulation, although the tangential field components also stimulate neurons [25]. Exciting the epileptogenic zone is unwanted, because that could make seizures easier to trigger. However, it is generally believed that tDCS is not strong enough by itself to elicit a seizure [26] but modulates the underlying brain excitability.

Most of the side effects of tDCS treatments are at the electrode level, such as skin itching, redness, and in extreme cases when the current is too high, burn marks on the skin. Additionally, slight dizziness has been reported. There are currently no other significant negative side effects reported from using tDCS on the brain [27].

In clinical practice, the procedure for applying tES is usually based on clinical experience rather than being extensively supported by electric field simulations. In most cases, standard stimulation parameters are applied in which the stimulation location, amplitude, and waveform are used as predefined by clinical protocols and the device manufacturer. For example, in [16], the use of long-term tES in the treatment of epilepsy is discussed. The stimulation focus was determined by visual inspection of the electroencephalogram (EEG), and large patch electrodes (5 × 7 cm and 5 × 5 cm) were used to approximately stimulate the focus. The shape of the human head and the electrical properties of the tissues in it vary significantly from person to person. Recent papers have suggested that taking this variability into account improves reaching the intended stimulation targets. In the work of Antonenko et al., a relation between the modeled field and the physiological effects of the stimulation is suggested [28]. Wagner et al. proved that when an accurate head model is used, containing at least isotropic skin, the skull, CSF, and brain gray and white matter, the standard electrode montage that is used in the clinic does not give a maximal electric field at the targeted location [29]. In two studies by Vorwerk et al. [30,31], the effects of the individual tissue parameters on the fields are explained, indicating the need for personalized models.

Building on that, recent studies have involved exploring the link between the simulated stimulation intensity and the clinical outcome [6,32]. These studies indicate that, by using standardized parameters for tDCS, there will be heterogeneity in the clinical outcome. In a recent study by Kaye et al. [7], a personalized location that was found by a multimodal presurgery evaluation was used to calculate the optimal stimulation montages, which led to good results. However, in this study, the head model was personalized only in patients with skull or brain defects. Our hypothesis is that focal brain disorders, such as epilepsy, can benefit effectively from personalized targeted focal stimulation of a sufficient intensity. To provide the means to answer that hypothesis, starting from the literature, we review the current techniques to personalize neurostimulation based on MRI images with the use of source maps from high-density (HD) EEG and possible personalized conductivity values.

## 3. Proposed Workflow

The proposed workflow from MRI and EEG to personalized tDCS can be represented in a flowchart, as shown in Figure 1. The different aspects and steps in the flowchart will be described in the following sections of the paper while putting them into perspective with the state of the art in the field. This has been achieved by analyzing the recent literature via, for example, PubMed searches and recent conference publications regarding neurostimulation. These searches indicate the growing interest and research in the field. The number of publications using tDCS as a keyword grew from 130 in 2010 to 962 in 2020 on PubMed and from 669 to 1945 on ScienceDirect over the same time range.

We start at the top of the flowchart with the two inputs to the workflow: EEG recordings and MRI input. The latter consists of structural images with the EEG electrodes co-registered. These will be used to get to the outputs in the form of source maps and a personalized tDCS model. The imaging input serves as the basis for the forward modeling problem, resulting in a leadfield matrix after several processing steps. These will be detailed in Section 4.1. The EEG input, together with the calculated leadfield matrix, allows for solving the inverse problem to create personalized source maps for a patient. The choices and approaches for the inverse problem will be described in Section 4.2. Using the results of the inverse modeling, together with forward modeling of stimulation electrodes, the induced currents and locations of electrodes for personalized stimulation can be determined from an optimization problem. This results in a personalized neurostimulation configuration or montage. Details on this approach will be provided in Section 5.2. Uncertainties, both in the modeling approach itself and in assumptions like the tissue parameters and electrode positioning, will influence the result of the optimization. Their effect will be reviewed in Section 6. The paper will be concluded with a discussion and an outlook in Section 7. Throughout the remainder of the review, numerous abbreviations appear. These are summarized in Table 1.

## 4. Determination of the Stimulation Target

### 4.1. The Forward Problem, Overview, and Techniques

#### 4.1.1. Segmenting the MRI Data

The first step for personalizing neurostimulation consists of segmenting the MRI data into different tissue types. This is a complex step [33] and should not be overlooked or oversimplified. The complexity lies in differentiating the tissue types that share unclear boundaries, especially in MRI images. This segmentation is usually performed with T1-weighted images, preferably with fat suppression. Taking the fat shift into account that happens in T1-weighted images certainly makes this problem non-trivial. When available, T2-weighted images should be used to improve the skull segmentation [34]. The results further down the modeling pipeline depend on this segmentation, which will be elaborated upon in Section 6.2.

Open source research packages for performing the segmentation are widely available, each based on different techniques and implementations. Freesurfer [35] and SPM [36] are used most commonly, although recent work using neural networks has proven to be faster and, in some cases, more accurate, especially when it comes to segmenting lesions and other abnormalities. This will be explained in more detail in Section 4.1.2.

The effects of segmentation errors or using a single generic head model can be quite severe. The skull is especially regarded to be a major factor in modeling inaccuracies [37]. A guideline for the list of tissues to be segmented and corresponding tissue properties was given by Vorwerk et al. [30,31]. Starting with the standard three-tissue model (brain, skull, and scalp), the authors showed that it is important to include cerebrospinal fluid (CSF) in the model due to its significantly higher conductivity than other tissues. Moreover, making the distinction between gray and white matter in segmentation has a significant impact on the modeling outcome. Modeling anisotropy was found to be of less importance. The inclusion of CSF is important when it can be correctly segmented. The exact conductivity of this highly conductive fluid is of less relevance due to the high contrast (difference in conductivity) with the rest of the tissues. A recent review on the effects of the quality of the segmentation was performed in [38]. In our review, where we only considered brain tissues, the most significant effects were found in the segmentation quality of the CSF.

#### 4.1.2. Handling Abnormalities

Structural abnormalities in the brain, such as tumors, lesions, skull defects, or artificial structures, should be taken into account during segmentation when accurate results are desired. One of the most straightforward but time-consuming methods is to manually correct the segmentation by using open source programs like ITK-SNAP [39]. This approach is time consuming and is still prone to human errors. These manual corrections, however, cannot be avoided when artificial structures are on or in the human head. Especially in the context of epilepsy, brain surgery and functional mapping with intra-cranial electrodes are common procedures, each resulting in non-standard structures in the human head model. These surgeries can lead to skull defects in the form of burr holes or discontinuities in the skull, which has been investigated by Datta et al. [40]. The subdural electrodes that are used to record the electrical activity on the cortical surface are usually attached to a polymer sheet with a very low conductivity, effectively blocking the currents between the brain and the scalp. Especially when a subdural grid of electrodes is used and the polymer sheet is large, this has to be taken into account for proper forward modeling [41,42].

Recent work on using artificial intelligence in the context of tissue segmentation has made the automatic segmentation of, for example, lesions or tumors viable. Hirsch et al. showed that by using knowledge about the spatial distribution of tissues, a deep volumetric neural network can segment MRI data with lesions equally as well as a clinical expert can [43]. Advances have also been made in segmenting and anatomically labeling the parts of the brain, in a similar way to how Freesurfer does, by using neural networks. This can reduce the processing time for these advanced segmentation types from hours to minutes [44]. From the mentioned research, it has become clear that artificial intelligence can aid segmentation in the case of abnormalities like lesions and tumors. For man-made abnormalities like holes or implanted electrode grids, manual work will always remain necessary. Good segmentation is usually performed only once, and further processing steps depend on it, so investing time in high-quality segmentation is advisable.

#### 4.1.3. The Generators of the EEG Signal

For modeling the translation from brain activity to the EEG signal, it is necessary to define which mechanism causes the signals at the EEG electrodes. One neuron by itself does not induce a sufficiently strong signal at the electrodes. It is generally assumed that patches of synchronized pyramidal neurons generate the measured signals. Without signal averaging, approximately six centimeters of cortical tissue containing these pyramidal neurons has to be synchronously active to generate a recordable potential on the scalp EEG [45]. Relative to other neurons in the brain, these pyramidal neurons are long and oriented normally to the cortical surface. They can thus be modeled as dipoles that are placed normal to the cortex [46]. Although the pyramidal neuron model for EEG is widely used, there are still open questions and problems with this model, which were summarized by Cohen [47].

#### 4.1.4. From Segmentation to the Head Model: the Methods

Based on segmentation, a voxelized computational electromagnetic model can be constructed in which the tissue types can be assigned to individual voxels (small cubic cells). The forward problem in EEG source localization is then equal to finding the projections of all possible sources, which are modeled as small dipoles in the head, to the electrodes [48]. In essence, the following three aspects are key to solving this problem: a head model, the electrode locations, and the tissue electrical conductivity values.

The desired accuracy and calculation time will dictate the choice of head model for solving the forward problem, together with the available data. In the absence of MRI data, one could use a spherical shell model or standardized head segmentation with standardized electrode locations to obtain a crude approximation of the reality. When individual MRI images are available, a personalized head model can be made. The EEG electrodes can be mapped via standard locations (with some slight manual adjustments) or by using a co-registration system, of which a few examples are given in [49].

When the locations of the scalp electrodes are known, the possible source points have to be determined. This can be a regularly spaced grid confined by the skull. The grid spacing determines the amount of points and thus the possible accuracy and computation time for the forward problem. The last choice that has to be made for the forward model is the solution method that will be used. In practice, three different solution methods are commonly used. These are summarized in Table 2 and will be explained next. In Figure 2, examples of the three different forward models are shown. The finite element method (FEM) model is shown using three and five different tissue types to illustrate the difference in the level of detail. Usually, the skin, skull, and a homogenized brain are used for a three-layer model, which can be refined by including CSF and by dividing the brain into white and gray matter.

For the spherical shell model, concentric spheres can be used to make a coarse approximation of the geometry, with layers of CSF, skull, skin and brain. For every source point in the model, it is possible to have an analytic solution for the potential on the shell surfaces. In this way, the contributions of all source points to the scalp potential at the EEG electrodes can be determined. The matrix that describes this linear mapping of the source points to the EEG electrodes is called the leadfield matrix. In general, due to the quasi-static assumption in EEG, there are no travel times for the signals. Propagation is assumed to be instantaneous.

A boundary element method (BEM) model, like a spherical shell model, works with nested surfaces. These surfaces are not necessarily spheres and can provide a geometrically accurate representation of the head shape. The downsides of such a model are the absence of analytical solutions and that the potentials on the shells have to be approximated numerically to determine the leadfield matrix.

FEM does not require nested surfaces. Instead, it subdivides the volume of the model into hexahedra or tetrahedra. Each element can have different material properties (conductivity) assigned, which can even be anisotropic (in contrast to the other two methods). Anisotropy will be explained in the next subsection. Finally, for FEM, the forward problem is solved over the finite elements to obtain the leadfield matrix. When the sensor locations and the head are both modeled accurately enough, the leadfield matrix is mainly dependent on the tissue properties.

#### 4.1.5. Tissue Parameters

Multiple tissue types are present inside the human head. The electrical conductivity of the tissues determines the propagation of electrical current densities. Since neurostimulation is a low-frequency application, conductivity of the tissue is the dominant parameter. Different tissue types have strongly different conductivity values, providing larger or smaller contrasts for the current densities at tissue transitions.

Tissues can also be anisotropic, which means that the material properties are direction dependent. The skull, for example, consists of conductive spongy bone (diploë) sandwiched between two layers of compact bone which are less conductive. This causes the conductivity in the directions tangential to the skull to be higher than the conductivity normal to it. White matter also exhibits anisotropic behavior, mainly due to the directions of the axons of the neurons [50]. The impact of this will be further explained in Section 6.

#### 4.1.6. Recent Trends

Recent trends for the forward problem boil down to improving the solvers in speed as well as accuracy. For FEM, work has been undertaken to implement a discontinuous Galerkin method, which has been proven to conserve charges and, in terms of accuracy, is superior to a continuous Galerkin implementation [51].

Recent work on solving the problem using the BEM has resulted in the implementation of a fast multipole method. This method makes it feasible to conduct accurate forward modeling for intracranial recordings and allows for speeding up the calculations. Still, it is impossible to use anisotropic material parameters in this method [52]. Another active topic in forward modeling is the inclusion of more tissue types in an attempt to further improve the quality of the solution. Recent examples include, for example, blood vessels [53] or the meninges [54]. The effect of adding more tissue types is usually small and refines the models. In Section 6, the effect of adding more tissue types will be explained in more detail.

On the segmentation side, there have been advances in skull segmentation by using, for example, zero-echo-time MRI sequences [55]. This technique, with which the skull can be more accurately segmented, requires extra MRI hardware, which makes this less feasible for a general workflow.

### 4.2. Inverse Problems, Overview, and Algorithms

#### 4.2.1. Inverse Solutions for Epilepsy

Source localization is a relevant and important topic in focal epilepsy and is still actively researched. In the work of Heers et al. [56], the accuracy of the source localization of interictal spikes is compared to the localization results from intracranial EEG. In the paper of Acar et al. [42], the same comparison is made, with the addition of modeling the plastic sheet of the intracranial electrodes. Two reviews of multiple studies and the correlation with intracranial EEG and surgery outcomes are given in [57,58], proving the relevance of electrical source localization in epilepsy. A very visual and clinical review of the analysis steps is given in [59]. Source imaging is usually performed on interictal spikes [60,61] because they are prevalent between seizures. Source imaging at seizure onset is more difficult and is investigated in [62]. In the next subsections, we will go into the fundamentals of inverse problems and source localization and give a more profound mathematical description.

#### 4.2.2. Inverse Problem Fundamentals

The forward problem was defined as the linear mapping between source points in the brain and recordings at the electrode level. The inverse problem follows quite naturally from that. Given the EEG recordings and the sensor locations, can we determine the sources that generated those recorded signals?

The head model usually contains a lot of nodes or voxels required for accurate representation of the segmented model. Therefore, the number of source points is much higher than the number of measurement electrodes. This imbalance makes the system inherently underdetermined and thus ill-posed, implying that there is an infinite number of solutions [63]. Using more electrodes (so-called high-density EEG) helps with the resolution that can be obtained, though the differences in the obtained solutions become negligible above a certain number (approximately 128) of electrodes [59,64].

By means of regularization, specific (more realistic) choices for solutions can be enforced, such as a solution with just a few activated neuron patches with respect to whole-brain activation. This pre-selecting can be performed by choosing one of the numerous source localization algorithms. This will be explained in more detail next.

Since the forward problem is defined based on a finite number of source points, the inverse problem can be written in matrix form as follows:(1)M=GD+n,
where M denotes the measured EEG potentials, G is the leadfield matrix, D is the dipole magnitude matrix, and n is the noise vector. At each source point, a dipole with three Cartesian amplitude components can be modeled. For this reason, the leadfield matrix will have a size N×3P, with *N* being the number of measurement electrodes and *P* being the number of possible source points.

With the source points on the surface of the cortex, following the shape of the sulci, the leadfield matrix can be reduced to a size N×P under the assumption that only the dipole component normal to the cortex contributes. This reasoning has an electrophysiological basis, because it assumes that only the pyramidal neurons are responsible for the EEG signal, as explained in Section 4.1.3. The measurement and source vectors can also be extended to be time-varying, although due to the quasi-static assumption of the problem, time does not influence the lead field matrix. Thus any delay from the source to the measurement electrodes is assumed to be zero.

The leadfield matrix G is calculated for the forward problem discussed in Section 4.1. At each defined source point, a dipole with a unit amplitude is simulated. The dipole magnitude vector D then represents the amplitude of these source points. The inverse problem can then be stated as follows: find a suitable approximation $\hat{D}$ for the dipole magnitude vector D in Equation (Equation 1). In the next subsection, the common methods for solving this problem will be explained.

#### 4.2.3. The Algorithms for Inverse Problems

Numerous algorithms have been developed over the years to solve this dipole magnitude matrix $\hat{D}$ while coping with the ill-conditioned nature of the problem. The main concepts for the inverse methods can be found in Table 3, which serve as the basis for more customized or specific algorithms, for which the solutions are given in Table 4. The relation between the concepts in Table 3 and the solutions in Table 4 can be seen in the methods family tree depicted in Figure 3. The overview is based on comparisons between algorithms and reviews found in the literature [65,66,67,68,69]. Each method is referenced in the table entries. The overall characteristics of the main algorithms is that they try to minimize Equation (Equation 1) with respect to a norm, with different options for regularization, using parameter λ. In the more specific algorithms (see Table 4), W denotes an arbitrary weighting matrix, B is a specific weighting matrix based on the columns of G, Δ is the discretized Laplacian operator, $r_{j}$ represents the electrode locations, $r_{\mathrm{dip}}$ represents the dipole locations, and C is the data co-variance matrix.

The minimum norm (MN) solution is the simplest solution and will minimize the 2-norm of the solution, giving the solution with the lowest amplitude and most spread. This is not a good solution if the source is very focal. It also introduces a depth bias, since the sources with the lowest amplitudes can be expected at the cortical surface. This depth bias can be overcome by introducing a certain weighting function to the solution, as performed in the weighted minimum norm and LORETA. sLORETA also weighs the solution of the minimum norm approach, but this algorithm takes the noise co-variance into account as well. The beamformer approach is the only approach discussed here that uses the properties of the data to make an estimate of the inverse operator. The single-dipole method fits the best single dipole to the data and thus gives the most focal solution, even if the underlying source is not. For this single dipole, six parameters have to be estimated, the location $r_{\mathrm{dip}}$ and the orientation, which is implied in D. These last two algorithms are part of a class of methods called parametric methods. Where the other algorithms find an equivalent current density representing the source, parametric methods find an equivalent current dipole. These algorithms usually require grid searches and iterations to find an optimal set of dipoles representing the source. Other methods in this family are, for example, the music and the nonlinear least squares methods [67].

Using these algorithms, source maps can be obtained, which can be used as input for the stimulation maps in the next section. A single-dipole source solution, for example, can directly denote a single stimulation location. In the context of epilepsy, when an epileptic zone has been found by source localization, this zone can be stimulated with cathodal stimulation to inhibit that brain area. In most patients, there is a unique, unilateral seizure onset zone despite possible bilateral interictal spikes. This is the main hypothesis in treating epilepsy via tDCS [16,26] and serves as the hypothesis in our work as well. Multifocal epilepsy is out of the scope of our current work.

#### 4.2.4. Accuracy of the Methods

Pascual-Marqui [68] found that the minimum norm estimate (MNE) and the linearly constrained minimum variance (LCMV) beamformer underperformed with respect to sLORETA. sLORETA exhibits zero localization error in a noiseless case, which is, however, far too idealized to be true in practice. sLORETA is one of the most used source localization methods in practice and has also been validated using clinical data [62]. Disadvantages of methods like sLORETA and MNE are that they give quite a spread in the source and they rely on regularization parameters. Specific research has been conducted on source localization on epileptic spikes, where methods that give sparser results in the source space are also used, like multiple sparse priors [61] (MSP), wavelet maximum entropy on the mean [70] (wMEM), and the focal under-determined system solver [71]. These methods have shown promising results, and the wMEM and MSP methods have been compared in a clinical setting where the computed source was compared to the resected brain tissue. The downside to these methods is that they are usually available in research software only [72] and thus are not often used outside of research.

#### 4.2.5. Software Implementations

Numerous software packages can be used to solve the forward and inverse problem, and a few of the open source packages are Brainstorm [73], EEGLAB [74], MNE [75], and Fieldtrip [76]. Their functionalities were compared to commercial software in [72], showing promising results. These software packages have been used extensively in research. MNE, for example, was used by Mikulan et al. to produce a dataset where stimulation has been applied in the brain using stereotactically implemented electrodes. Simultaneous high-density EEG was recorded to provide a ground truth model for source localization [77].

#### 4.2.6. Recent Trends

The algorithms in Table 4 have been integrated into clinical practice. Some alternative techniques have been investigated for the algorithms. For instance, Awan et al. [78] described approaches using neural networks and maximum entropy on the mean (MEM).

Furthermore, the authors of [79] investigated whether a combination of EEG-fMRI and EEG source localization could be used to obtain results with the spatial resolution of fMRI and the temporal resolution of EEG. They specifically investigated this for epilepsy patients. While this study found a mismatch in the estimated spatial location of the epileptic zone, another study performed by Centeno et al. improved this method and obtained better results [80]. They also compared the combined localization to the clinically obtained epileptic zone and predicted surgery outcomes for a small patient population. The predictive performance was shown to be above the chance level.

### 4.3. Recommendation for Workflow

In the case of focal cortical epilepsy, which is the target population for further studies, the preferred source localization should be conducted with dipole fitting, given that enough EEG electrodes are used. A dipole fit gives a direct target with an orientation, which can be used as the input in the montage optimization routine in our workflow.

## 5. Personalizing tDCS Electrode Montages for Optimal Target Stimulation

### 5.1. Software Implementations

There are commercial options for electromagnetic simulation software available that can perform calculations for neurostimulation, such as Sim4Life [81]. This is a general purpose simulation suite for biomedical applications ranging from thermal and MRI simulations to electromagnetics-induced neuronal activation models. Alternatively, open source software packages have become available, which are more tailored to specific biomedical applications. For instance, for neurostimulation purposes only, there is SIMNIBS [82] and ROAST [83]. A comparison between these packages has been presented in [84]. The main differences in the results between those two packages were found to be caused by differences in segmentation by the two packages.

The focus will be on the SIMNIBS software package, since this can incorporate the workflow proposed (see Figure 1) in a very natural and intuitive way. This package can be used to find the optimal stimulation parameters, given a stimulation focus, while taking into account constraints on the maximum total current, the amount of electrodes, and the current per electrode.

It consists of a set of tools and functions that can be used to go from MRI data to electric field maps, optimized electrode locations, electric field uncertainty maps, and much more. The tool has a graphical user interface (GUI) but can be called from MATLAB [85] or Python [86] as well, making scripting straightforward.

### 5.2. Optimization of Electrode Locations and Currents

When a stimulation target in the brain is known (i.e., defined by the source localization techniques described earlier), an optimized montage can be calculated, which consists of the optimal electrode locations and currents. By taking the individual head geometries into account in the modeling software, the dose that each patient receives can be controlled in such a way that the maximum field will always be the same (e.g., a certain field strength at the target) [87]. In this way, studies on the effect of tDCS can be compared much better. The optimization procedure is not straightforward and involves several trade-offs. The most important trade-off is in selecting which figure of merit based on the electric field should be optimized. This can either be the maximum amplitude or the focality [88]. In the case of maximizing the field in the target, this usually leads to a reciprocity type of solution [89]. These solutions are obtained by solving the forward problem from the region of interest and using the same excitation on the electrodes as is obtained with the forward model. The other figure of merit, focality, involves making the field as focal as possible. This corresponds to a variant of a least squares solution, in which the figure of merit is the integral over the electric field in the ROI divided by the integral of the electric field outside the ROI. This gives a measure of focality of the solution.

Another trade-off relates to the number of electrodes used. To optimize an electrode montage, usually a leadfield matrix is constructed in which a lot of electrodes are registered to the scalp and the field of each electrode is calculated. Based on the leadfield matrix and a constraint on the amount of electrodes, the montage can be optimized. Instead of a maximum number of electrodes, a sparse optimization or a minimum current threshold can be used to limit the amount of electrodes. An example of two optimizations for the same location with the same amount of total current can be found in Figure 4. In Figure 4a, the optimal patch solution is given, with blue denoting the negative current (cathode) and red the positive current (anode). In Figure 4b, an example of an HD optimization is given where a maximum of 10 active electrodes was set, with red denoting the positive current, blue the negative current, and green no current. On the cortex, the norm of the electric field is plotted, with the highest values in red. The color scale for the electric field is the same for both optimizations.

Little is known about which regions not to stimulate. When the stimulator is placed close to the forehead, the eyes may need special attention to avoid retinal stimulation and to avoid inducing phosphenes. This can be taken into account using weight factors on the desired fields in the cortex.

### 5.3. Trends in tES Stimulation and Simulation

One of the biggest future trends in the field of computational modeling for neurostimulation lies in the direction of multi-scale models, where macro field simulations and neuron level activation models can be combined to explain and predict the working of neuromodulation on a deeper level [90]. Such simulations are difficult to perform at a large scale due to the computational complexity of simulating enormous amounts of single neurons. On a targeting level, temporal interference is being investigated as an alternative for reaching deeper targets like in DBS and as an improvement on tDCS. In this technique, currents are applied through electrode pairs at different locations. These pairs are excited at slightly different frequencies (10–20-Hz difference). This will create a localized interference pattern deeper in the brain (below the cortex), which may cause neuronal activation at the interference site [91].

## 6. Effects of Model Uncertainties and Noise

The computational models that are made for simulating source imaging or neurostimulation are rarely fully in agreement with the measurements. This can be attributed to the model uncertainties that are inherently present. The uncertainties in MRI segmentation were already explained in Section 4.1. In this section, other relevant uncertainties that relate to modeling will be explored, and recent work on quantifying them will be discussed.

### 6.1. Uncertainties Due to Electrical Noise

Mosher et al. derived a Cramér–Rao bound for the intrinsic accuracy limit present in source localization using a four-shell model to obtain analytical formulas [92]. This Cramér–Rao bound is independent of the algorithm and gives the minimal possible error standard deviation for unbiased estimators. In this paper, the authors proved that averaging the EEG signal over multiple timeframes decreases the minimum obtainable error. A simulation study was performed by Whittingstall et al., in which it was found that the difference in localization error, due to electrical noise, is minimal for a tangential or a normal source with respect to the cortex, even for sources deep in the brain [93].

### 6.2. Specific Forward Model Errors

Song et al. investigated the effects of the MRI voxel size on the source localization. To accomplish this, they conducted resampling from 0.88 × 0.88 × 0.9 mm voxels to 0.5 × 0.5 × 0.5 mm, 1 × 1 × 1 mm and 2 × 2 × 2 mm voxels [94]. They found that, for the coarse sampling of 2 × 2 × 2 mm, the skull deformed significantly, and the localization error, with respect to the ground truth, was in the order of 10 mm for a cortical source. This error was found for 1 × 1 × 1 mm as well, but the reason why was not explained. The resampled finer voxels led to no improvement in image quality but provided a denser grid for the solver. There was no source localization error in that case. Akalin Acar et al. discussed the effect of specific forward modeling errors on the inverse problem [95]. They found that using the MNI template, which is a commonly used standardized human head made by averaging 152 MRI registrations [96], introduced a median localization error of 5 mm. They recommended always using an individual head model. The error that was created by using spherical shells for the forward method gave the same magnitude of error as using the MNI template. As mentioned in Section 4.1, in the study by Vorwerk et al. [30], a six-compartment model was constructed, including compact as well as spongy bone, CSF, gray matter, and anisotropic white matter. They used this model as a reference solution to compare the forward solutions of less refined models. From that approach, they found which tissues and properties contributed the most in accurate forward modeling. The addition of CSF and the gray and white matter distinction to the standard three-layer scalp–skull–brain model was found to be required. The influence of adding the anisotropic behavior of the white matter was found to be of less significance. A specific study on how to model the skull was reported by Montes-Restrepo et al. [97]. In this study, they compared CT-based skull segmentations and MR-based skull segmentations and modeled them as a layered isotropic (thus containing the diploë), a single-layer anisotropic, and a single-layer isotropic. They found that when CT images were available, the layered isotropic model gave the best results with regard to dipole localization error. When only MR images were used, which resulted in lower quality segmentation, it was difficult to generate the layered model. In that case, the isotropic single layer gave better results than the anisotropic case. They attributed this to the complex skull anatomy.

### 6.3. Tissue Conductivity Uncertainties

A derivation for the sensitivity of EEG recordings for tissue conductivity variations based on the quasistatic Maxwell equations and an FEM approach is given in [98]. This sensitivity can be visualized at each possible source to obtain an impression for the expected error for different source locations. Radich et al. derived the Cramér–Rao bound for dipole location and gave an analytic expression for a four-shell spherical model [99]. Saturnino et al. used generalized polynomial chaos to implement efficient mapping from probability density functions for tissue parameters to probability densities in the resulting electric field during stimulation [100].

In the clinic, standard database values for electrical conductivity are routinely used in calculations for stimulation or source imaging. However, there are multiple methods to assess the tissue conductivities, which significantly improve the accuracy of the simulations when taken into account. In the work of McCann et al. [101], a review is given on the spread of tissue conductivities with different measurement techniques. An overview of these methods is listed in Table 5, with a brief description of the essential features per method. The most elementary method is resecting a part of the tissue and finding the conductivity by, for example, the four-probe method [102]. The problems with this method are that it is invasive and destructive and that the properties of the resected tissues might change due to the resection. The other listed methods in Table 5 measure the conductivity in an indirect way, and three of them (electrical impedance tomography (EIT), E/MEG, and magnetic resonance electrical impedance tomography (MREIT)) also rely on solving an inverse problem. The signal-to-noise level that is achievable at the moment, together with other factors like the magnetic field induced by the cables in EIT measurements, make these methods currently not reliable enough to implement in the clinic. Diffusion tensor imaging (DTI) is suited to quantify the anisotropy in tissues but not to quantify the absolute conductivities in the tissues. In [101], a relation is found between certain tissue conductivities, like the skull and age. This was further investigated in [103], in which the effect of this age dependency on the tDCS fields is underlined. There are several more reasons for variability in tissue conductivities, such as the measurement method used, the tissue temperature, and inter-individual variations. For accurate source reconstruction, the skull conductivity was found to play a significant role [95]. Modeling this individually could be very powerful in making accurate models. In a recent study by Schrader et al., it was proposed to use the P20/N20 response to calibrate the skull conductivity in the clinic. This P20/N20 response is a response to a specific somatosensory stimulation paradigm for which the generator location is known. A combined EEG and MEG source analysis on the evoked potential could be used [104]. The downsides of this technique are that the other conductivities have to be assumed and that the analysis has been conducted on a four-layer sphere model with isotropic conductivities. Translating this to realistic head models will thus require more research.

### 6.4. Electrode Registration Uncertainties

Another difference between the models and the clinical practice is the placement of both the measurement (recording) and stimulation electrodes. The placement will rarely be perfect, especially in home applications. For the inverse problem, Beltrachini et al. used the algorithm-independent Cramér–Rao bound to conclude that the error due to the deviation in electrode position, which was assumed to be 5 mm for a 120-electrode montage, was negligible with respect to the error caused by electrical noise [106]. For tES, Opitz et al. recommended that, based on modeling and intracranial measurements, the electrode placement error should be less than 1 cm to obtain reliable results in congruence with the measurements [41].

### 6.5. The Influence of the Uncertainties on the Workflow

Obtaining an accurate result from the workflow proposed starts with high-quality data. The combination of a T1- and T2-weighted scan gives a superior segmentation quality with respect to a T1-weighted image only. Proper electrode placement for EEG, as well as filtering and preconditioning, was not treated in this paper, although they are very relevant topics [59]. Since the segmentation and construction of the head model has to be performed just once for each patient, one should always choose for the highest accuracy possible, given that memory is not a big issue for such models. Five-tissue FEM implementation is thus preferable, since it also models the CSF, which has been proven to have a significant impact when added to the model. Ideally, one would have a model of the tissue conductivities, either by direct or indirect measurements. When such methods prove to be reliable, accurate, and easily applicable in a clinical setting, they should definitely be taken into account. Knowledge on the EEG electrode positions at a millimeter resolution is not fully necessary, but modeling them with a position error lower than 1 cm is advised and can be accomplished with digitization systems or photos of the montage.

## 7. Discussion and Outlook

In this work, we have presented a workflow and essential ingredients for personalization of tDCS, which is based on knowledge of individual electric field distributions that can potentially lead to improved clinical efficiency of tDCS in focal epilepsy patients. Although metrics like field strength and focality show promise for this, we have also identified knowledge gaps and shortcomings that still require work to make tDCS a viable treatment in the clinic.

While this paper was written for neurostimulation to treat a single disease, it might be possible to treat multiple diseases at the same time. In the case of epilepsy, a specific co-morbidity is depression. Fregni et al. described how treating one of the two might relieve the other disease as well [107]. With more focal stimulation and HD tDCS montages, it might be feasible to stimulate at multiple places at the same time. Such optimizations could be seamlessly integrated in the workflow presented here.

In addition to personalizing the simulation of tES to maximize and control the effect of tES for a certain patient, for therapeutic interventions, it will be important to be able to administer the stimulation outside the hospital setting as well. Making such stimulation devices portable and wearable is not trivial [108], especially when real-time seizure probability prediction is to be implemented [109,110]. Personalizing these at-home treatments and seizure probability predictions is, in that case, a must due to the heterogeneity in anatomy and disease.

Finally, although this paper was written based on finding and stimulating the epileptic zone, it is not the only way in which epilepsy can be treated. Research on epileptic networks in the brain, using network metrics, has shown promising results. These network analysis tools have been embedded in open source software as well [111]. Promising results have also been found in stimulating networks, together with the target, to enhance the response [112].

From the abundance of literature, it is clear that the field of neurostimulation is rapidly expanding. Novel clinical results can be explained by recent advancements in in vivo measurements and electric field modeling, opening the door for reliable personalized neurostimulation.

We propose that tDCS montages can be personalized using EEG data and structural MRI data. These personalized tDCS protocols could lead to higher clinical efficacy of tDCS treatment in epilepsy patients. Even though the focus here was on the treatment of epilepsy patients, the methods described here could be translated to other pathologies with a clearly defined focal origin (i.e., stimulation target).

Stimulation parameters like the stimulation duration, number of stimulation sessions, and total current (or expected field strength at the target) were not discussed in this paper. These parameters are, at the moment, mostly part of a universal protocol, and further research on their clinical effects has to be performed. Personalizing these parameters in an automatic way using measurement data may prove to be difficult. Future trends in improving the segmentation quality and tissue conductivity estimation will make the workflow an even more viable clinical tool.

The workflow we presented here provides the key aspect to improving the relation between simulation metrics and clinical results. Applying this in clinical trials will further help develop the method.

## Figures and Tables

**Figure 1 brainsci-12-00610-f001:**
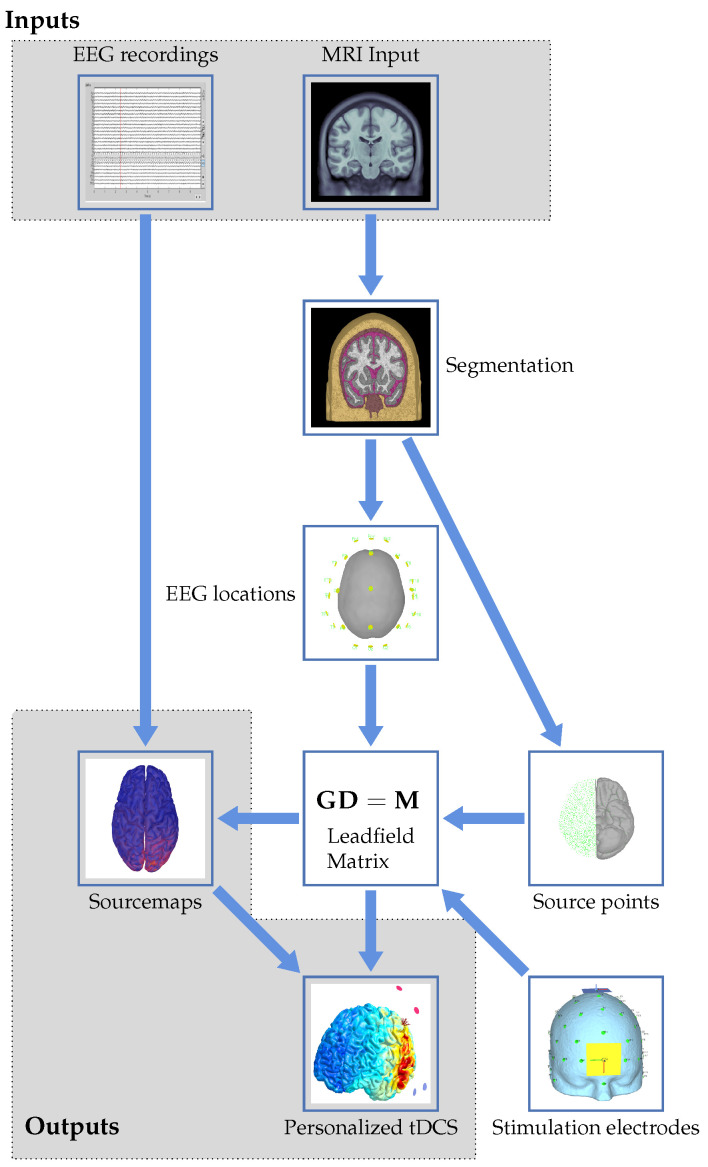
Graphical flowchart showing the proposed workflow to arrive at personalized tDCS from EEG and MRI data.

**Figure 2 brainsci-12-00610-f002:**
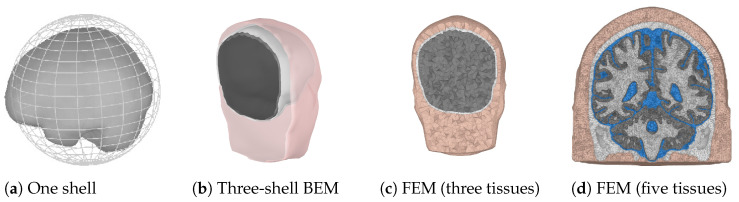
Examples of three different forward models. Colors are used to indicate different tissue types.(**a**) A one sphere brain representation, with the brain surface indicated for completeness. (**b**) Pink represents the skin shell, white the outer skull shell, and black the inner skull shell. (**c**,**d**) Pink and white are used for the skin and the skull, respectively. Gray is used for the whole brain in (**c**). Gray and white are used for the gray and white matter, and blue is used for the CSF in (**d**).

**Figure 3 brainsci-12-00610-f003:**
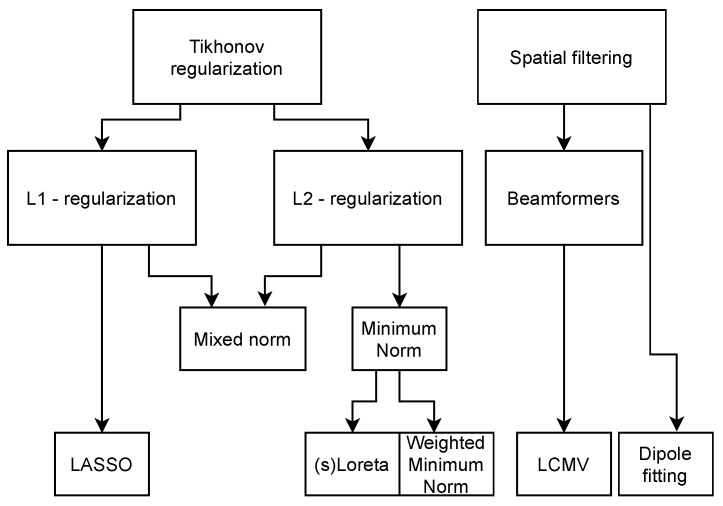
Inverse methods family tree, indicating their interdependency.

**Figure 4 brainsci-12-00610-f004:**
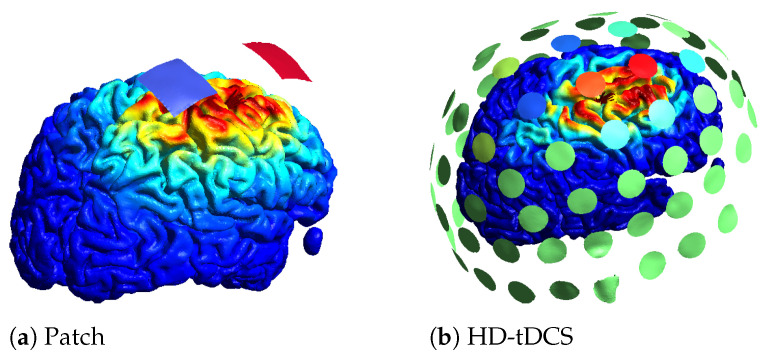
Different optimization results obtained using SIMNIBS.

**Table 1 brainsci-12-00610-t001:** List of abbreviations used.

Abbreviation	Definition
AED	anti-epileptic drugs
BEM	boundary element method
CSF	cerebrospinal fluid
DBS	deep brain stimulation
DTI	diffusion tensor imaging
EEG	electroencephalogram
EIT	electrical impedance tomography
FEM	finite element method
MEM	maximum entropy on the mean
MNE	minimum norm estimate
MRI	magnetic resonance imaging
MSP	multiple sparse priors
ROI	region of interest
tACS	transcranial alternating current stimulation
tDCS	transcranial direct current stimulation
tES	transcranial electric stimulation
TMS	transcranial magnetic stimulation

**Table 2 brainsci-12-00610-t002:** Forward problem solution methods, based on [48].

Method	Pros	Cons
Spherical shells	-Analytical solution -Fastest method	-Low accuracy -Idealized geometry
Boundary element method (BEM)	-Fewer mesh points required -Fastest calculations	-Low accuracy -No anisotropy
Finite element method (FEM)	-Most accurate -Freedom in meshing -Can handle anisotropy	-Large matrix, iterative solvers -Lead field poorly conditioned

**Table 3 brainsci-12-00610-t003:** Inverse method fundamentals.

Inverse Method	Formulation
Tikhonov regularization [65]	$\hat{D}=\min_{D}\;||GD-{M||}^{2}+\lambda L\left(D\right)$
Ridge (L2) regression [66]	$\hat{D}=\min_{D}\;||GD-{M||}_{2}^{2}+\lambda {\left|D\right|}_{2}$
Lasso (L1) Regression [66]	$\hat{D}=\min_{D}\;||GD-{M||}_{2}^{2}+\lambda {\left|D\right|}_{1}$

**Table 4 brainsci-12-00610-t004:** Inverse method specific algorithms.

Inverse Method	Formulation
Minimum norm [67]	$\hat{D}={\left(G^{T}G+\lambda I\right)}^{-1}\;G^{T}M$
Weighted MN [67]	$\hat{D}={\left(G^{T}G+\lambda W^{T}W\right)}^{-1}\;G^{T}M$
LORETA [67]	$\hat{D}={\left(G^{T}G+\lambda B\Delta^{T}\Delta B\right)}^{-1}\;G^{T}M$
sLORETA [68]	$\hat{D}=D_{\mathrm{MN}}^{T}s_{jj}^{2}D_{\mathrm{MN}}$
LCMV beamformer [69]	$\hat{D}\left(r\right)=$ ${\left[G{\left(r\right)}^{T}\cdot C^{-1}\cdot G\left(r\right)\right]}^{-1}G{\left(r\right)}^{T}C^{-1}\cdot M$
Single dipole [67]	$\hat{D}=\min_{r_{\mathrm{dip}},D}\;||G\left(r_{j},r_{\mathrm{dip}}\right)D-M\left|\right|$

**Table 5 brainsci-12-00610-t005:** Measurement techniques for conductivity estimation.

Method	Description
Directly Applied Current (DAC)	Invasive method Current injected Voltage measured
Electrical Impedance Tomography (EIT) [105]	Noninvasive method AC current applied Potential differences recorded Inverse modeling to find conductivity map
E/MEG	Noninvasive method Source localization Optimize conductivity and match measurements Relatable to EIT
Magnetic Resonance EIT (MREIT)	Noninvasive method Apply currents to body in MRI Measure magnetic flux density Inverse modeling to find conductivity map
Diffusion Tensor Imaging (DTI)	Noninvasive method Diffusion-weighted MRI Acquire diffusion tensors Construct conductivity maps

## Data Availability

Not applicable.

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
