# Peer review of "Personalized tDCS for Focal Epilepsy—A Narrative Review: A Data-Driven Workflow Based on Imaging and EEG Data"

_brainsci, 2022, doi:10.3390/brainsci12050610_

Round 1

Reviewer 1 Report

The manuscript entitled “Personalized tDCS for focal epilepsy: a data-driven workflow based on imaging and EEG data” by Beumer et al, is a very well-writen manuscript and covers most of the important methodological points in tDCS. However, I have some comments that can clarify several points.

1.- The manuscript is categorized as Article, but I’m not convinced of this. In fact, it seems more a review (by the length too) because there are not primary data at all. I read the manuscript looking for those data but I did not find. I think that the title is some misleading too, because the authors should indicate that this is more a review (although in a form of proposal).

2.- The manuscript has a lot of abbreviations. I think that it would benefit from a table.

3.- The references 45 and 46 are quite strange. The former is about qEEG and the latter is an EEG atlas when the authors are speaking about biophysical and physiological topics. Please, maybe references more accurate would be appreciated (e.g Nunez, Shrinivasan or Niedermeyer, Lopes-da-Silva or whatever else they prefer).

4.- In several places it’s stated that the inclusion of more tissue types improves the quality of the solution (e.g, lines 296, 508 or 511). However, the sources of EEG are mainly exclusive at cortex. Why the white matter anisotropy could improve the solution from cortical sources?.

5.- Please, explain the meaning of regularization (line 327).

6.- I guess that cortical dipoles are of the same magnitude (p) for every cortical segment (voxel). However, probably this magnitude would be higher during the interictal spike (when a greater amount of neurons are synchronized) than during the physiological processing for the same region. How do this difference in p would affect to the model?.

7.- In line 336 it is said that “only de dipole component normal to the cortex contributes”. Are you excluding the pyramidal neurons placed into the sulci?.

8.- In line 377 it is said that “when an epileptic zone has been found (…) can be stimulated”. However, as previously the authors affirmed, usually what can be identified is the irritative zone (indicated by the presence of interictal activity). How do this hypothesis fits with the reticular theory of epilepsy?. For example, in temporal lobe epilepsy it’s usual the presence of bilateral spikes, so, do you need to stimulate in both temporal lobes?.

9.- Although acronyms in line 537 are indicated at table 4, maybe they must be defined at text.

Reviewer 2 Report

At the manuscript " Personalized tDCS for focal epilepsy: a data-driven workflow based on imaging and EEG data" by Dr. Steven Beumer et al it was analyzed the relevant literature for the processing data of conventional transcranial electric stimulation (tES). Authors presented a processing workflow to personalize tES for focal epilepsy, for which there is a clear cortical target to stimulate.

Authors also identified shortcomings and ongoing trends with regards to, segmentation quality and tissue conductivity measurements. Authors believed that presented processing steps result in personalized tES based on metrics like focality and field strength, which allow for correlation with clinical results.

Very impressive review, I have no significant complaints

Minor criticism:

  1. Figure 2 legend does not explain exactly what the colors in part B mean.
  2. Perhaps add a color scale in Fig.4?

The presentation of a subject is systematic and comprehensive, list of references is quite full. 

Round 2

Reviewer 1 Report

I have no more questions.